# Comparison of a SARS-CoV-2 mRNA booster immunization containing additional antigens to a spike-based mRNA vaccine against Omicron BA.5 infection in hACE2 mice

Jacklyn R. Hurst[1], Maedeh Naghibosadat[1], Patrick Budowski[2], Jun Liu[3], Philip Samaan[4], Frans Budiman[5], Alexandra Kurtesi[6], Fredo Qi[6], Haritha Menon[3], Rajesh Krishnan[3], Jumai Abioye[3], Anne-Claude Gingras[6,7], Mario Ostrowski[5], Natalia Martin Orozco[3], Robert A. Kozak[1,4,8]*

1 Biological Sciences Platform, Sunnybrook Research Institute at Sunnybrook Health Sciences Centre, Toronto, ON, Canada, 2 Institute of Medical Sciences, University of Toronto, Ontario, Canada, 3 Providence Therapeutics Holdings, Inc., Calgary, AB, Canada, 4 Department of Laboratory Medicine and Pathobiology, University of Toronto, Ontario, Canada, 5 Department of Medicine, University of Toronto, Toronto, ON, Canada, 6 Lunenfeld-Tanenbaum Research Institute at Mount Sinai Hospital, Sinai Health System, Toronto, ON, Canada, 7 Department of Molecular Genetics, University of Toronto, Toronto, ON, Canada, 8 Division of Microbiology, Sunnybrook Health Sciences Centre, Department of Laboratory Medicine and Molecular Diagnostics, Toronto, ON, Canada

* rob.kozak@sunnybrook.ca

**Data Availability Statement:** All relevant data are within the paper and its Supporting Information.

## Abstract

The emergence of SARS-CoV-2 variants presents challenges to vaccine effectiveness, underlining the necessity for next-generation vaccines with multiple antigens beyond the spike protein. Here, we investigated a multiantigenic booster containing spike and a chimeric construct composed of nucleoprotein (N) and membrane (M) proteins, comparing its efficacy to a spike-only booster against Omicron BA.5 in K18-hACE2 mice. Initially, mice were primed and boosted with Beta (B.1.351) spike-only mRNA, showing strong spike-specific T cell responses and neutralizing antibodies, albeit with limited cross-neutralization to Omicron variants. Subsequently, a spike-NM multiantigenic vaccine was then examined as a second booster dose for protection in hACE2-transgenic mice. Mice receiving either homologous spike-only or heterologous spike-NM booster had nearly complete inhibition of infectious virus shedding in oral swabs and reduced viral burdens in both lung and nasal tissues following BA.5 challenge. Examination of lung pathology further revealed that both spike-only and spike-NM boosters provided comparable protection against inflammatory infiltrates and fibrosis. Moreover, the spike-NM booster demonstrated neutralization efficacy in a pseudovirus assay against Wuhan-Hu-1, Beta, and Omicron variants akin to the spike-only booster. These findings indicate that supplementing spike with additional SARS-CoV-2 targets in a booster immunization confers equivalent immunity and protection against Omicron BA.5. This work highlights a promising strategy for individuals previously vaccinated with spike-only vaccines, potentially offering enhanced protection against emerging coronaviruses.

**Funding:** This work was primarily supported by a grant from the Canadian Institutes of Health Research (CIHR). Some reagents were supplied by Providence Therapeutics, including the vaccines which were the focus of this study. CIHR had no role in study design, data collection and analysis, decision to publish, or preparation of the manuscript. Providence Therapeutics provided minimal input in the study design limited to issues related to vaccine assessment, storage, dosage and delivery.

**Competing interests:** Providence Therapeutics provided minimal input in the study design limited to issues related to vaccine assessment, storage, dosage and delivery. This does not alter our adherence to PLOS ONE policies on sharing data and materials.

## Introduction

The SARS-CoV-2 pandemic has caused more than 772 million cases and nearly 7 million deaths as of December 19th, 2023 (https://covid19.who.int). Several intramuscular vaccines have been approved and are currently used worldwide that utilize the transmembrane spike glycoprotein as the immunogen. The spike protein is the main target of host-generated neutralizing antibodies (nAbs) that inhibit SARS-CoV-2 infection by blocking the binding of the viral receptor binding domain (RBD) to angiotensin-converting enzyme 2 (ACE2) on host cells and preventing fusion of the viral envelope with the cell membrane [1]. While the spike protein is the key protective determinant with 90% or more of neutralizing activity dependent on RBD-binding antibodies in convalescent sera, and up to 99% neutralizing activity in vaccinated sera [2, 3], the spike protein continues to evolve to evade both convalescent and vaccine-elicited immunity by facilitating virus escape from nAbs [4–8]. The changes in the spike proteins of recent Omicron lineages represent a continuing challenge for current vaccine strategies and have resulted in compromised immunity in vaccinated individuals, symptomatic breakthrough infections and increased transmission potential in the community [9–15]. Additionally, there is an ongoing risk of other coronaviruses (CoV) with pandemic potential (e.g. MERS-CoV) and the potential for new CoVs to emerge.

To overcome the loss of vaccine efficacy and combat immunological escape, bivalent or multivalent vaccines are a key strategy to broaden immunogenicity against currently circulating variants, as well as increase the breadth of neutralization against future emerging variants. The urgent need to expand protection against circulating Omicron variations has led to the authorization of variant-specific bivalent vaccinations; [16, 17], but whether variant-specific boosters elicit superior neutralizing antibody responses against emerging variants than the original monovalent vaccine is debated. The WT/BA.1 bivalent vaccine showed a modest 1.7-fold higher pseudovirus-neutralizing antibody (nAb) titers against BA.5 compared to the original mRNA-1273 [16], and the WT/BA.5 bivalent vaccines have shown only 1.2 to 1.3-fold higher pseudovirus-nAb titers against BA.5 than their original WT vaccines [18, 19]. Furthermore, while the BA.5 bivalent booster can expand humoral responses against Omicron XBB subvariants in relatively healthy individuals [20, 21], it largely fails to protect against XBB.1 and XBB.1.5 in a target group of immune-deficient cancer patients [22]. In response to declining vaccine effectiveness and the emergence of variants with increased infectivity and immune evasion, universal (pan-SARS-CoV-2) vaccines that contain multiple conserved virus antigens may be a better strategy for robust, broad, and durable immunity and are an active area of research [23, 24]. Notably, protective immune responses have been shown with vaccines expressing other SARS-CoV-2 antigens such as nucleocapsid (N), membrane (M) and envelope (E) proteins [25–27]. These viral proteins exhibit higher sequence conservation than spike, have indispensable roles in virus replication [28, 29], and have been shown to induce robust T cell immunity [30–34]. High frequencies of multi-cytokine-producing CD8+ T cell responses specific to the SARS-CoV-2 N and M proteins have been associated with milder disease [35–38] and cross-reactivity with select seasonal coronaviruses [39], suggesting the importance of including non-spike proteins for future vaccine designs that target both seasonal and SARS coronaviruses.

The first step towards a pan-SARS-CoV-2 vaccine is to confirm that the inclusion of additional SARS-CoV-2 antigens in a booster vaccine does not result in inferior immunity or reduced protection against severe disease compared to the current routine vaccinations containing spike only. Thus, in the present study we investigate whether a multivalent booster that contains spike, nucleoprotein, and membrane protein can provide similar protection as a spike-only booster against the previously prevailing Omicron BA.5 variant in

hACE2-transgenic mice and examine the effect on spike protein neutralization to other variants of concern. The vaccination series included monovalent Beta (B.1.351) spike alone as an intramuscular prime/boost series, as this is representative of the general population where a majority of people have received a full prime and booster series against the SARS-CoV-2 spike protein, followed by a second booster 2 months later containing either B.1.351 spike alone or a 1:1 mix of mRNAs encoding B.1.351 spike and nucleoprotein-membrane protein chimera (spike-NM).

## Methods

### Vaccine formulations

The spike mRNA vaccine used in these studies encode the full-length spike from Beta B.1.351 that does not contain the 2P stabilization. The spike-NM mRNA vaccine is a 1:1 mass ratio mixture of the B.1351 spike mRNA and an mRNA encoding a chimera of regions of the nucleotide and membrane proteins of the SARS-CoV-2 Wuhan-Hu-1 strain. All mRNAs used were modified mRNA using N1-methyl-pseudouridine and formulated in lipid nanoparticles as described previously [40].

### Animals and viruses

Six-to-eight-week-old female K18-hACE2 transgenic C57BL/6 mice were purchased from Charles River. Animals were housed at the BSL2 facility at the University of Toronto for vaccinations and transferred to the BSL3 facility at least 72 hours prior to challenge experiments to allow for proper acclimatization. In both facilities, mice were housed under specific pathogen-free conditions with food and water ad. All animal work described in this study were performed in compliance with the Canadian Council on Animal Care guidelines and approved by the Animal Care Committees of The University of Toronto (APR-00005433-v0002-0). All personnel were specifically trained for handling mice in the BSL3 facility. The virus used for animal challenge studies was passage 1 of SARS-CoV-2 BA.5 (BEI Resources NR56798) which was grown in CaLu-3 cells (ATCC HTB-55). The viral RNA was submitted to TACG Facilities at SickKids Hospital (Toronto, Ontario) for sequencing to ensure the resulting viral stock was representative of the circulating variant.

### Immunization and challenge studies

The spike mRNA vaccine used in these studies encode the full-length spike from Beta B.1.351 that does not contain the 2P stabilization. [40], and thus, a 10 μg dose of the spike-only mRNA vaccine was utilized for this study. K18-ACE2 transgenic mice were intramuscularly injected with monovalent B.1.351 spike mRNA (10 μg) in a total volume of 50 μL per dose with either mRNA diluted in formulation buffer or formulation buffer alone (sham vaccination). All mice received a boost intramuscular vaccination 3 weeks later with the same treatment as dose 1. Mice received vaccinations under isoflurane anesthesia and were monitored daily following vaccination for clinical signs of distress including lethargy, ruffled fur, accelerated breathing, and hunched posture. Each indication was scored on a scale of 0–3 per category and if animals reached a score of 3 for an individual parameter or a cumulative score of 7 at any time after vaccination, they would be humanely euthanized immediately. For immunogenicity experiments, the planned endpoint was 3 weeks following the boost vaccination, with mice ($n \geq 4$ per group) euthanized by terminal blood collection and cervical dislocation. Blood and spleens were collected for the immunogenicity endpoint. No animals were euthanized before the planned endpoint.

For challenge experiments, mice received a third dose of vaccination that contained either monovalent B.1.351 spike mRNA alone (10 μg), or multiantigenic mRNA containing 5 μg of B.1.351 spike mRNA and 5 μg of a chimeric of nucleoprotein and membrane protein mRNA (spike B.1.351 +NM), or formulation buffer as a control 8 weeks after the boost vaccination (week 11). The amount of spike in the heterologous spike-NM booster was decreased to 5 μg to maintain the overall dose of the spike-NM mRNA vaccine at 10 μg. Eight weeks following the second booster immunization (week 19), all mice were challenged intranasally with SARS-CoV-2/BA.5 (dose 1 x $10^6$ $TCID_{50}$) under isoflurane anaesthesia. Mice were monitored daily to evaluate clinical indications of suffering and disease, including weight loss, lethargy, accelerated respiration, posture, and ruffled fur, each scored on a scale of 0–3 per category. If animals reached a score of 3 for an individual parameter, a cumulative score of 7, or excess weight loss exceeding 20% at any time after challenge, immediate endpoint euthanasia of the animal would be required to alleviate animal suffering. Oropharyngeal swabs were collected at 1- and 3-days post-infection (dpi) in PBS under anesthesia for measurement of viral shedding. Experimental endpoint after challenge was defined at 5 days post-infection, and mice were euthanized by terminal bleeding and cervical dislocation under isoflurane anesthesia with blood, lungs, and nasal turbinate tissues collected. No animals were euthanized before the planned endpoint. The total duration of the experiment from the first vaccination to endpoint was 19 weeks plus 5 days as shown in Fig 4.

## ELISpot assay

To determine the number of cells secreting mouse IFN-γ and IL-4, mouse IFN-γ ELISPOT[PLUS] and mouse IL-4 ELISPOT[PLUS] kits containing ELISpot plates pre-coated with rat anti-mouse IFN-γ (AN18) and rat anti-mouse IL-4 (11B11) monoclonal antibodies respectively, were purchased from Mabtech, Cincinnati, OH. On the day of the experiment, plates were washed 4 times with phosphate buffered saline (PBS, Gibco[TM]) and blocked with 200 μl RPMI-10 medium (RPMI-1640 supplemented with 10% FBS, 100U penicillin, 100 μg streptomycin, and 2mM L-glutamine, all purchased from Invitrogen) for at least 1 hour. Splenocytes were added onto the plates at 2.5 x $10^5$ splenocytes per well and stimulated with SARS-CoV-2 B.1.351 spike peptide pools at 1 μg/ml/peptide. The spike peptide pools were purchased from JPT Peptide Technologies GmbH, Berlin, Germany and scanned the entire spike glycoprotein of the B.1.351 lineage; containing a pool of 315 peptides derived from 15-mer peptides with 11 amino acid overlaps. The same volume of 40% DMSO (Sigma-Aldrich), the solution to dissolve each peptide pool, was used as the negative control. PMA/Ionomycin (Sigma-Aldrich) was used as the positive control. After 18 hours of incubation in 37˚C, 5% $CO_2$, cells were removed, and plates were washed 5 times with PBS. Biotinylated detection anti-IFN-γ and anti-IL-4 antibodies were then added to their respective plates and incubated at room temperature for 2 hours. Following incubation, plates were washed with PBS and Streptavidin-horseradish peroxidase (HRP) enzyme conjugate (Mabtech) was added onto plates for 1 hour at room temperature. Plates were then washed again and room temperature TMB ELISpot substrate (Mabtech) was added onto plates and left until spots were visible. Spot development was stopped by washing extensively with Milli-Q water and left to dry overnight in the dark. The number of spots were acquired with an ImmunoSpot® Analyzer (Cellular Technology Limited, Cleveland, OH) and quantified by subtracting the number of the spots of the DMSO control wells from the number of the spots of the corresponding peptide pool stimulation wells. Results were expressed as spot-forming units (SFU) per 2.5 x $10^5$ splenocytes.

## Flow cytometry

Mouse splenocytes were cultured in RPMI-10 and stimulated with the B.1.351 spike peptide pool at 1 μg/ml/peptide in the presence of GolgiStop^TM and GolgiPlug^TM (BD) for 6 hours. 40% DMSO and PMA/Ionomycin were used as the negative and positive controls respectively. Cells were first stained with the LIVE/DEAD^TM Fixable Violet Dead Cell Stain, blocked the FcR with the TruStain FcX (Biolegend, San Diego, CA) and then stained with the fluorochrome-labeled anti-mouse CD3/CD4/CD8/CD44/CD62L mAbs (all purchased from Biolegend except CD44 from BD). Cells were then treated with Cytofix/Cytoperm (BD) and stained with fluorochrome-labeled anti-mouse IFN-γ/TNF-α/IL-2/IL-4/IL-5 mAbs (Biolegend). LSRFortessa was used to acquire the flow cytometry data, which were then analyzed with FlowJo. Percentage of cytokine+ T cells was calculated by subtracting the percentage of the DMSO control cells from the percentage of the corresponding peptide pool stimulation cells.

## Quantification of infectious SARS-CoV-2 titer

VeroE6 cells cultured in complete DMEM (supplemented with 100U/mL penicillin streptomycin and 10% FBS, DMEM-10) were seeded into 96-well plates and incubated overnight at 37˚C. On the following day, culture medium was removed and serial 10-fold dilutions of samples in DMEM with 2% FBS (DMEM-2) were added onto the cell monolayer. For collected tissues, tissues were weighed and homogenized in serum free DMEM with stainless steel beads on a bead mill prior to serial dilutions. After overlaying samples, plates were incubated at 37˚C for 1 hour with gentle shaking every 15 minutes. Following incubation, infection media was replaced with 100 μL DMEM-2 per well, and cells were left to incubate for up to 10 days at 37˚C. Cytopathic effect (CPE) was checked on day 7 and up to day 10 where applicable. $TCID_{50}$ (50% tissue culture infectious dose) was defined as the highest dilution factor of the inoculum that yielded 50% of the cells with CPE and determined by using the Spearman-Karber $TCID_{50}$ method. Tissue sample infectious titers were normalized to the weight of the tissue.

## RT-qPCR

Molecular quantification of viral loads was completed using real-time (RT)-PCR to determine the genomic copies of SARS-CoV-2 in oropharyngeal swabs and tissue homogenates and was performed according to the published protocol [41]. For oropharyngeal swabs, swabs were eluted in 500 μl PBS and RNA was extracted using QIAamp Viral RNA Mini kit (QIAGEN, Toronto, ON, Canada). For collected tissues, tissues were weighed and homogenized with stainless steel beads (Qiagen) in 600 μl Buffer RLT and RNA was isolated from tissue homogenates according to the RNeasy Mini kit (QIAGEN), including the QIAshredder spin column step. Luna Universal Probe One-step RT-qPCR kit (New England Biolabs, Ipswich, MA) was used to amplify the envelope (E) gene using the Rotorgene Q according to methods previously described [42] using the following primers and probes: forward primer: `ACAGGTACGTTAATAGTTAATAGCGT`, reverse primer: `ATATTGCAGCAGTACGCACACA`, and probe CAL Fluor Orange `560- ACACTAGCCATCCTTACTGCGCTTCG-BHQ-1`. The cycling conditions were 1 cycle at 60˚C for 10 minutes, then 95˚C for 2 minutes, followed by 40 cycles at 95˚C for 10 seconds and 60˚C for 15 seconds. To generate standard curves, dilutions of a synthetic plasmid containing a segment of the E gene were used (GenScript, Piscataway, NJ) and run at the same time for conversion of Ct value to genomic copies.

## Pseudovirus neutralization assays

The spiked-pseudotyped lentivirus neutralization assays were performed to assess neutralization capacity as previously described [40, 43]. Briefly, lentiviral virus-like particles were generated from co-transfection of the viral packaging (psPAX2, Addgene, #12260), the ZsGreen and luciferase reporter construct (pHAGE-CMV-Luc2-IRES-ZsGreen-W, provided by Jesse Bloom), and the spike protein constructs: Wuhan-Hu-1 (D614G), Beta, Omicron BA.1, BA.2, BA.2.12.1, BA.5, and XBB.1.5 subvariants into HEK293TN cells (System Biosciences [LV900A-1]). Mouse sera samples were serially diluted (two-fold for the testing of XBB.1.5 and four-fold for all other pseudoviruses) and incubated with each pseudovirus at a 1:1 ratio for 1 hour at 37˚C. The final dilution of mouse sera is 1:40 to 1:5120 for XBB.1.5, and 1:80 to 1:1310720 for all other pseudoviruses. The sera-virus mixture was then transferred to plated HEK293T-ACE2/TMPRSS2 cells and incubated for an additional 48 hours at 37˚C. Cells were lysed, and Bright Glo luciferase reagent (Promega, Madison, WI) was added for 2 minutes. Luciferase signals were measured with a PerkinElmer Envision instrument (PerkinElmer, Waltham, MA). To estimate the 50% neutralization titer ($ID_{50}$), the luciferase signals were first normalized to positive and negative controls, and then fit to a four-parameter logistic model using the drm function in the drc package in R [44]. The lower limit of quantitation (LLOQ) is defined as the starting dilution of the sera (1:40 for XBB.1.5 and 1:80 for all other pseudoviruses). Samples that did not show neutralization capacity at the serum dilutions tested were assigned an ID50 value of 1.

## Histopathology

Tissues were fixed in 10% neutral phosphate-buffered formalin, routinely processed, sectioned at 5 μm, and stained with hematoxylin and eosin (H&E) for histopathologic examination. Lung tissues were examined for the presence or absence of features of cell or tissue damage as well as regeneration and repair. Masson's trichrome staining was also performed to assess tissue architecture and gauge the progression of fibrosis. Inflammation and fibrosis were assessed by an observer blinded to identity of the mouse group. Images were acquired on Tissue Scope LE (Huron Digital Pathology, Waterloo, Canada) under the 20x objective. The region of aggregation of inflammatory cell infiltrates is delineated and represented as a percentage of the cell infiltrates area to total tissue area. To quantify fibrosis and an automated pipeline was used for the processing of Masson Trichrome (MT). A counting algorithm was used to assess blue pixel and red pixel density The regions of collagen content stained in blue were delineated and represented as percentage fibrosis area to total tissue area.

## Statistical analysis

All figures were generated using Prism 10.2.2 (GraphPad Software Inc.). Statistical analyses were performed using Mann Whitney's unpaired t test, one-way ANOVA using Holm-Šídák's, Tukey's, or Dunn's multiple comparisons test, or two-way ANOVA using Tukey's, or Šidák's, or Dunnett's multiple comparisons test with a single pooled variance. *P* values are shown were applicable and considered significant if $p < 0.05$. Raw data is provided the S1-S16 Tables in S1 File.

# Results

## SARS-CoV-2 mRNA vaccine candidates

The coding sequences of the full-length S (amino acids 1 to 1273) were based on the spike protein from the SARS-CoV-2 B.1.351 isolate (Fig 1A). The NM chimeric construct encodes a

**(A)**

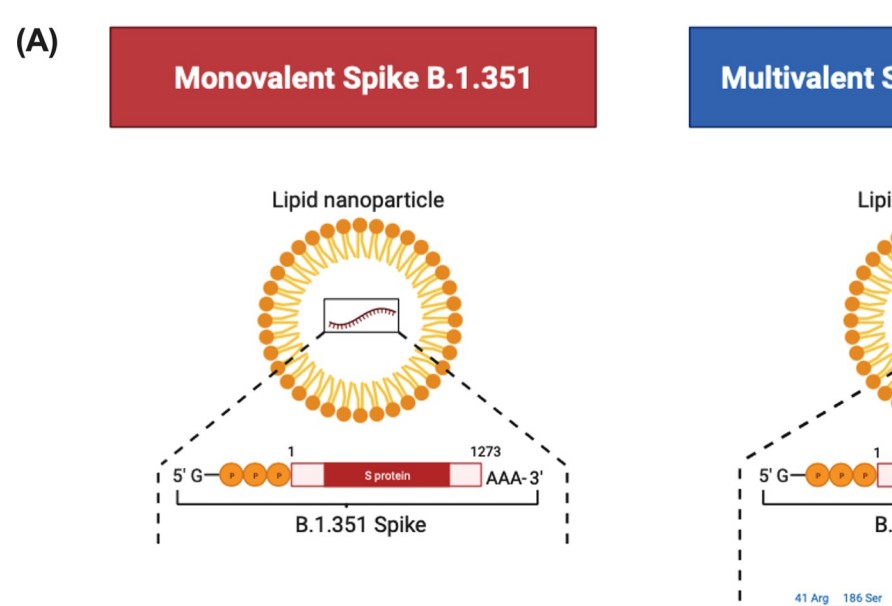

**(B)**

| Immunization Treatments | Dose 1 | Dose 2 | Dose 3 |
|---|---|---|---|
| Group 1 (Spike alone) | Spike B.1.351 | Spike B.1.351 | Spike B.1.351 |
| Group 2 (NM Boost) | Spike B.1.351 | Spike B.1.351 | Spike B.1.351 **+ NM** |
| Group 3 (Control) | Sham | Sham | Sham |

**(C)**

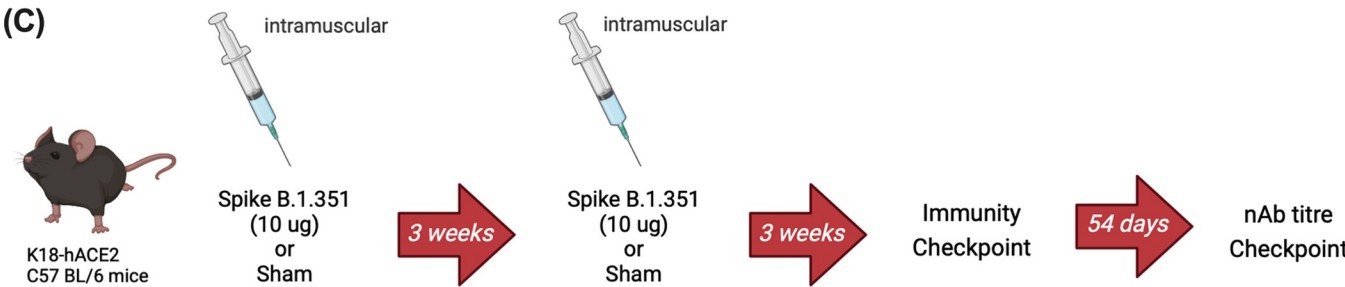

**Fig 1. Vaccine and treatment group details. (A)** Vaccine constructs schematic. Spike B.1.351, SARS-CoV-2 Beta spike mRNA (S, amino acids 1–1273). Spike B.1.351 + NM, 1:1 mixture containing SARS-CoV-2 Beta spike mRNA (S, amino acids 1–1273), and SARS-CoV-2 Wuhan-Hu-1 nucleoprotein (N), and membrane protein mRNA (M). **(B)** Immunization treatment groups. All vaccinated mice will receive two doses of monovalent spike B.1.351 mRNA, followed by either an additional monovalent spike B.1.351 boost or a multiantigenic spike B.1.351 + NM boost targeting the B.1.351 spike as well as the nucleoprotein (N) and membrane (M) protein. Control mice received formulation buffer only. **(C)** Schematic representing the vaccination regimen for the immunogenicity checkpoint. Six- to eight-week-old female C57BL/6 transgenic mice expressing human ACE2 receptors (K18-hACE2) were immunized intramuscularly twice with 50 μL containing 10 μg SARS-CoV-2 Spike B.1.351 mRNA or formulation buffer only at three weeks apart. Three weeks after the second immunization, mice were euthanized with blood and spleens collected to assess humoral and cell-mediated immune responses. Fifty-four days following the booster immunization, mice had saphenous bleeds drawn to assess the longevity of humoral immunity.

fusion protein of part of the Wuhan-Hu-1 nucleocapsid protein and part of the membrane protein (Fig 1A). The chosen regions of N and M protein are enriched in T cell epitopes reported for COVID19 convalescent patients in Toronto [45] and other parts of the world. The codons and sequence in the mRNA construct were codon optimized to increase the translation of the encoded proteins. A plasmid was produced, linearized, and purified to use as template in an in vitro transcription reaction for production of mRNA. Formulation of the lipid nanoparticle was performed using purified single mRNA of either Spike or NM, or combination of two mRNAs for B.1.351 +NM.

## Spike B.1.351 vaccination induces strong spike-specific cell-mediated and humoral immunity in K18-hACE2 mice

To confirm the immunogenicity of the spike B.1.351 mRNA construct, transgenic female K18-hACE2 C57BL/6 mice were intramuscularly vaccinated twice, 3 weeks apart with 10 μg of spike B.1.351 mRNA per dose formulated in lipid nanoparticles (Fig 1C). Control mice received the same volume of formulation buffer only. Three weeks following primary and booster immunizations, mice were humanely euthanized with spleens and serum collected to monitor cellular and humoral immunity respectively.

Three weeks after the second immunization, cell-mediated responses were evaluated by enzyme-linked immunospot (ELISpot) and flow cytometry to determine T cell immunogenicity. Splenocytes from individual mice ($n \geq 4$ per group) were stimulated with S peptide pools from the SARS-CoV-2 B.1.351 lineage to measure interferon-γ (IFN-γ) and interleukin-4 (IL-4) producing T cells by ELISpot. Spike-B.1.351 vaccinations induced 838.75 ± 220.37 (means ± SEM) IFN-γ spot-forming units (SFU) per $2.5 \times 10^5$ splenocytes and had very few IL-4 SFU above the background detected, confirming strong $T_H1$ responses to spike B.1.351 following vaccinations (Fig 2A). Furthermore, splenocytes were analyzed for cytokine producing $CD4^+$ and $CD8^+$ T cells by intracellular cytokine staining and flow cytometry following overnight stimulations with spike B.1.351 peptide pools. Both $CD4^+$ and $CD8^+$ T cells from vaccinated mice had increased percentages of IFN-γ, TNF-α, and IL-2–producing cells compared to control mice and very low percentages of IL-4– and IL-5–producing cells following stimulation with Beta spike, indicating a strong induction of a TH1 response to the spike protein. Interestingly, $CD8^+$ T cells from vaccinated mice had IFN-γ and TNF-α-producing cells of a higher percentage than that of $CD4^+$ T cells following Beta spike stimulation. Together, splenocyte stimulations show that B.1.351 spike vaccinations induce strong SARS-CoV-2 beta spike-specific T cell responses.

Humoral responses against pseudovirus containing B.1.351 spike were also measured at three weeks following the second immunization. As expected, monovalent Beta B.1.351 spike vaccinations elicited high nAb levels against the matched pseudovirus harbouring the Beta B.1.351 spike with a median nAb neutralization ($ID_{50}$) titer at 40401 (IQR, 29414 to 105853), while control mice receiving the formulation buffer only did not contain any neutralizing antibodies (Fig 2C). To confirm the durability of humoral responses, mice were bled again 8 weeks following the second immunization to measure nAb levels. Neutralizing titers against the pseudovirus harbouring the Beta B.1.351 spike persisted over time, with a median $ID_{50}$ titers of 47907 (IQR, 23022 to 104172), while control mice continued to show no nAb titers (Fig 2C).

## Spike B.1.351 only vaccine does not induce broadly neutralizing antibody titers in K18-hACE2 mice

Cross-neutralization of SARS-CoV-2 Wuhan-Hu-1 (D614G) and variants of concern (VOCs) were then determined using lentivirus particles pseudotyped to harbor the same mutations in

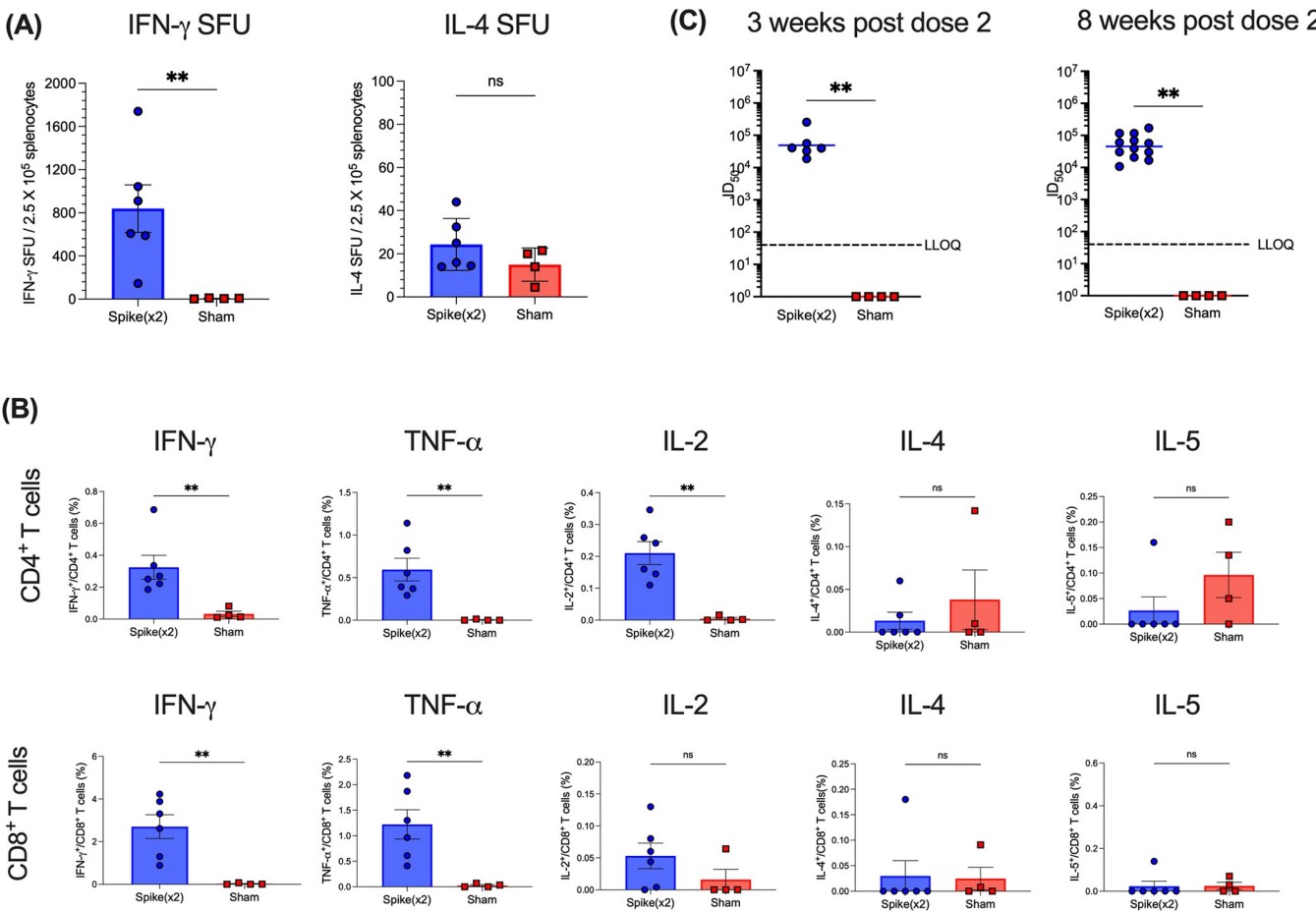

**Fig 2. Immunogenicity of spike B.1.351 in K18-hACE2 mice.** K18-hACE2 transgenic mice expressing human ACE2 receptors were immunized with SARS-CoV-2 spike B.1.351 mRNA or formulation buffer only for control mice. All mice received two intramuscular immunizations containing 10 μg of mRNA per dose, one on day 0 and another on day 21 (n ≥ 4 per group). Three weeks after the booster immunization, splenocyte cell suspensions from individual mice were stimulated with peptide pools encompassing the full Beta spike (S) glycoprotein protein from SARS-CoV-2. Control mice received formulation buffer only (sham). **(A)** IFN-γ and IL-4 cytokine production were detected by ELISpot. Data points represent spot forming units (SFU) per 2.5 X $10^5$ splenocytes from individual mice. **(B)** Percentage of cytokine-producing cells in CD4+ and CD8+ T cells were detected by intracellular cytokine staining and flow cytometry. Each symbol represents one mouse (n ≥ 4 per treatment group). Horizontal lines indicate the mean, and error bars represent standard error. Statistical comparisons were made using one-way ANOVA with Tukey's multiple comparisons (* $p \le 0.05$). **(C)** Serum neutralizing antibody titers from vaccinated mice 3 weeks and 8 weeks post dose 2 were analyzed against SARS-CoV-2 Beta (B.1.351) pseudovirus. Samples that did not neutralize viruses at the lowest serum dilution (1:80) are designated an $ID_{50}$ titer of 1. Dotted line represents the lower limit of quantitation (LLOQ). Statistical comparisons were made using Mann-Whitney unpaired t tests (** $p \le 0.01$).

the spike protein that are found in Wuhan-Hu-1 (D614G), Beta (B.1.351), and Omicron (BA.1, BA.2, BA.2.12.1, BA.5, and XBB1.5) strains using sera of mice at both 3 and 8 weeks following the second immunization. Vaccinated mice showed effective cross-neutralization of Wuhan-Hu-1 (D614G) with median $ID_{50}$ titers of 15820 (IQR, 8478 to 53641) at 3 weeks post-second dose, and these responses persisted to 37436 (IQR, 16305 to 61908) median $ID_{50}$ titers by 8 weeks post-second dose (Fig 3). At both 3 and 8 weeks after the second immunization, vaccinated mice exhibited 2- and 3-log-fold higher $ID_{50}$ titers against Omicron pseudoviruses BA.1, BA.2, BA.2.12.12.1, and BA.5, respectively, compared to control mice; however, these increases were not statistically different. Pseudoviruses bearing spike protein from Omicron XBB1.5 showed more resistance to neutralization than the other strains with nAb titers comparable to control mice at either time point following the second immunization. Control mice

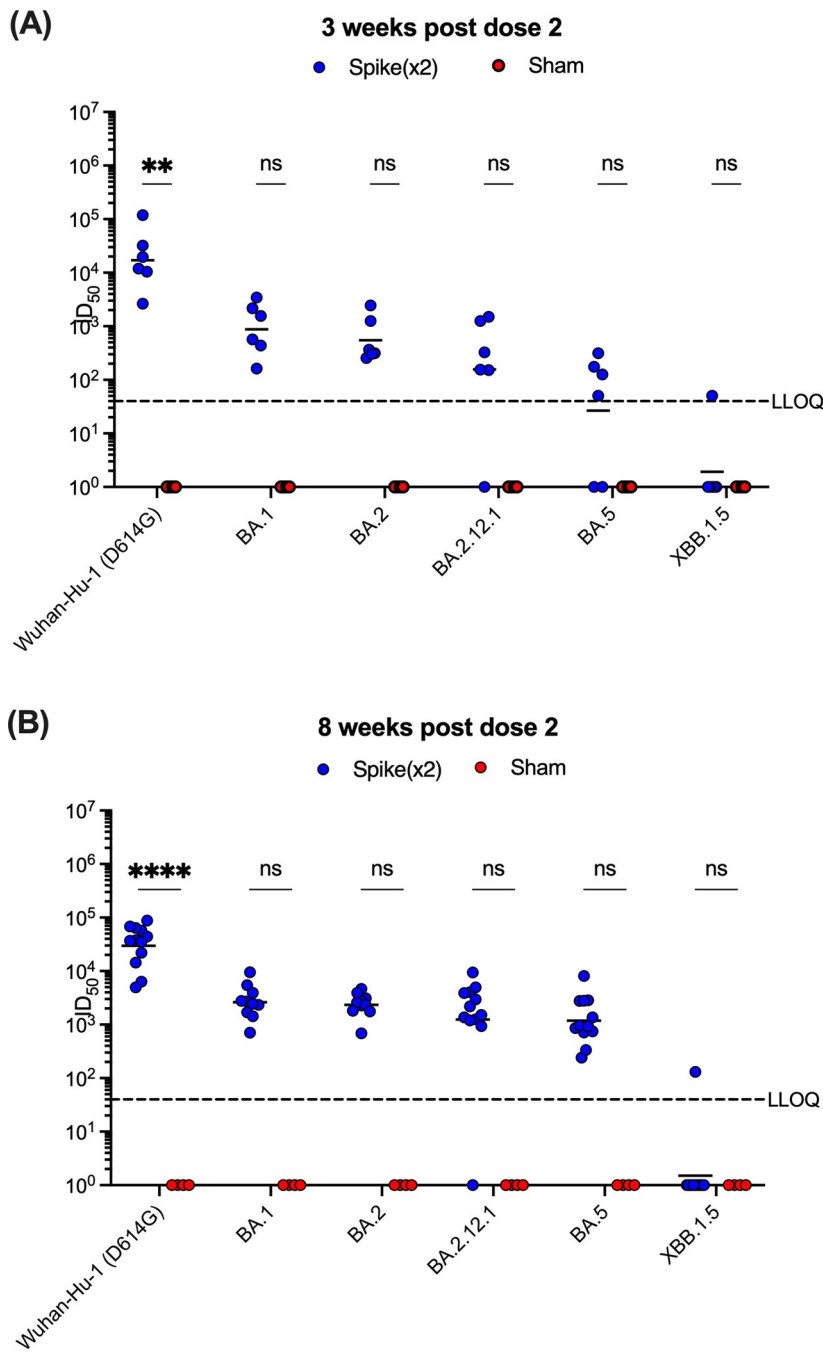

**Fig 3. Pseudovirus neutralization of SARS-CoV-2 VOCs from vaccinated mice sera.** Serum neutralizing antibody titers from vaccinated mice **(A)** 3 weeks and **(B)** 8 weeks post dose 2 were analyzed against Wuhan-Hu-1 (D614G), and Omicron (BA.1, BA.2, BA.2.12.1, BA.5, and XBB1.5) strains. Each symbol represents $ID_{50}$ titers one mouse ($n \geq 4$ per group). Samples that did not neutralize viruses at the lowest serum dilution (1:80 for XBB1.5, 1:40 for all other pseudoviruses) are designated an $ID_{50}$ titer of 1. The horizontal line indicates the median $ID_{50}$ titers for each group. The dotted line represents the lower limit of quantitation (LLOQ). Statistical comparisons were made using two-way ANOVA with Tukey's or Šidák's multiple comparisons test with a single pooled variance (***$p \leq 0.001$; ****$p \leq 0.0001$).

receiving the formulation buffer only did not have any detectable nAb titers in their sera. Together, these results demonstrate that the vaccine-matched Beta B.1.351 pseudovirus was the most sensitive to neutralization, and that cross-neutralization to Wuhan-Hu-1 (D614G) persisted over time. Vaccination with Beta (B.1.351) spike alone proved to exhibit reduced nAb activity against Omicron pseudoviruses, which led to the use of a multiantigenic vaccine as a booster dose before challenge.

## SARS-CoV-2 Omicron BA.5 challenge and viral shedding

We next sought to compare the effect of monovalent and multiantigenic booster immunizations on preventing SARS-CoV-2 Omicron BA.5 infection in K18-hACE2 transgenic mice. Following primary and booster immunizations with monovalent spike-only (10 μg), mice received a third intramuscular dose 8 weeks later containing either B.1.351 spike alone (10 μg), or half the amount of spike B.1.351 mRNA (5 μg) and 5 μg of a chimeric mRNA containing nucleocapsid protein and membrane protein. Control mice received immunizations with formulation buffer only. Three weeks following the third immunization, all mice were intranasally challenged with 50 μL containing $10^6$ $TCID_{50}$ SARS-CoV-2 BA.5 (Fig 4A). Mice were monitored daily to evaluate clinical indications of disease. When compared to pre-challenge baseline weights, vaccinated mice showed their greatest weight loss at 4 days post-infection (dpi), with median weight losses of 14.76% and 20.09% for monovalent and multiantigenic boosted mice respectively (Fig 4B). In contrast, control mice had their greatest weight loss at 5 dpi, with median weight loss of 19.57% (Fig 4B). Both groups of vaccinated mice recovered by 5 dpi, whereas the weights of control mice continued to trend downwards. Furthermore, control mice presented their highest scores of disease indications at 5 dpi, while both vaccinated groups had improved clinical scores by 5 dpi (Fig 4C).

Throughout infection, viral shedding was monitored in oral swabs for all groups by $TCID_{50}$ and by measuring spike envelope (E) gene amplification in real-time reverse transcription–polymerase chain reaction (RT-PCR) (Fig 5C and 5F). At 1 dpi, control mice showed 2-log-fold higher $TCID_{50}$ titers compared to monovalent and multiantigenic-boosted mice, and by 3 dpi virus shedding in oral swabs had depleted across all groups (Fig 5C). Significant reductions of SARS-CoV-2 E gene RNA copies were found in oral swabs from both vaccinated groups compared to control mice at both 1 and 3 dpi, with less viral RNA copies shown in monovalent-boosted mice compared to multiantigenic-boosted mice at 3 dpi, however, both vaccinated groups had significantly less viral RNA shedding compared to control mice at both time points (Fig 5B).

At 5 dpi, the viral burden was determined in the lungs and upper airways after euthanasia. Control mice contained significantly higher amounts of infectious virus in lung and nasal tissues compared to both groups of vaccinated mice (Fig 5C, 5E). Lower amounts of E gene RNA copies were also detected in lung tissues from both monovalent and multiantigenic-boosted mice compared to control mice, with 4.56 and 6.39 $log_{10}$ RNA copies, respectively, compared to control mice with 11.11 $log_{10}$ RNA copies, however, statistical significance was only achieved in the monovalent-boosted mice (Fig 5D). In nasal tissues, both monovalent and multiantigenic-boosted mice had significantly reduced viral RNA with 5.375 and 5.773 $log_{10}$ RNA copies, respectively. In comparison, control mice had 7.661 $log_{10}$ RNA copies (Fig 5F) in nasal tissues.

## Monovalent and multivalent booster immunizations prevent lung pathology by SARS-CoV-2 Omicron BA.5 challenge

The left lobes of lungs from all mice were used to compare lung pathology across treatment groups using a semiquantitative grading system to score pathology severity. Pathology was

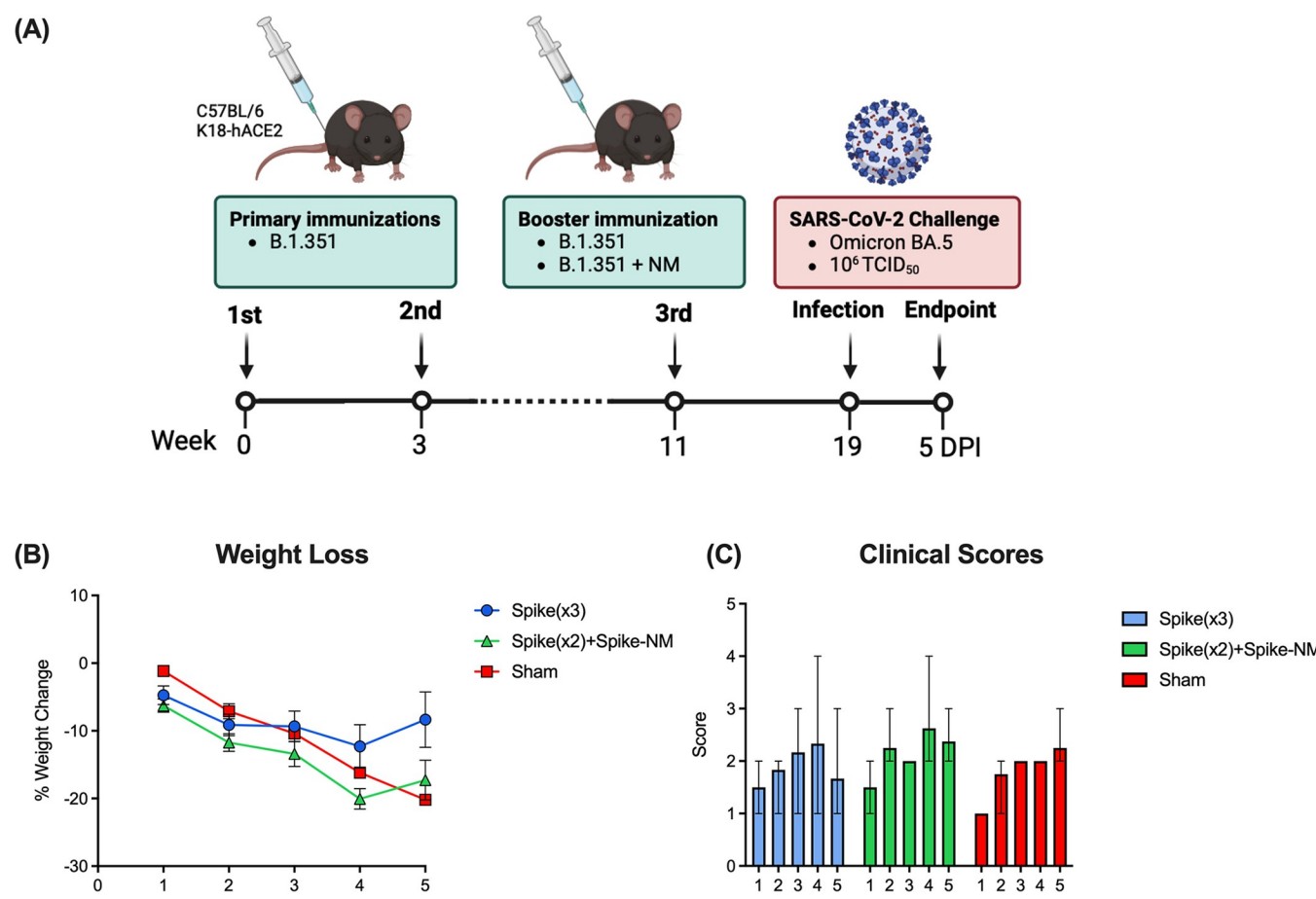

**Fig 4. Monovalent and multivalent booster vaccinations reduce weight loss in K18-hACE2 mice following SARS-CoV-2 Omicron BA.5 challenge. (A)**
Schematic representing vaccination and challenge regimen. Six- to eight-week-old female K18-hACE2 mice were intramuscularly immunized twice with
SARS-CoV-2 spike B.1.351 mRNA three weeks apart while control mice received a sham immunization containing formulation buffer only. Eight weeks after
the second immunization, mice were intramuscularly immunized with a third dose of either spike B.1.351 monovalent (10 μg) or the indicated multiantigenic
containing spike B.1.351 mRNA (5 μg) and a chimeric mRNA containing nucleocapsid protein and membrane protein (5 μg), or buffer only (control). Three
weeks following the second booster, mice were intranasally challenged with $10^6$ TCID$_{50}$ SARS-CoV-2 Omicron BA.5. Mice were monitored and scored daily
for weight loss and clinical signs of disease. **(B)** Weight loss. Data is represented as a percentage of day 0 weight. Data points represent the mean weight
loss ± SEM (n ≥ 4 per group). **(C)** Clinical scores following SARS-CoV-2 challenge were assessed on weight loss, respiration, lethargy, posture, and ruffled fur,
and were scored on a scale of 0–3 per category. Bars represent mean clinical scores per group with range (n ≥ 4 per group).

observed to be greater in the lungs of unvaccinated mice compared to lungs of mice that
received either of monovalent or multiantigenic booster immunizations. Unvaccinated mice
presented greater amounts of tissue damage and lesions in their lungs, characterised by bron-
cho epithelial necrosis (Fig 6A). Disease scores were significantly higher in the lungs of unvac-
cinated mice compared both groups of vaccinated mice but were comparable between the
monovalent and multiantigenic boosters. Both vaccinated groups had reduced scores for
inflammatory patterns compared to unvaccinated mice, which was comprised of necrosis,
intra-alveolar macrophages, and mononuclear inflammation around airways (Fig 6B). Fur-
thermore, both vaccinated groups showed lower percentages of fibrosis compared to unvacci-
nated mice, scored by lower collagen deposition (Fig 6C). Taken together, we conclude that
both spike-only and spike-NM boosters demonstrated comparable protection from severe
lung pathology at 5 dpi with SARS-CoV-2 Omicron BA.5.

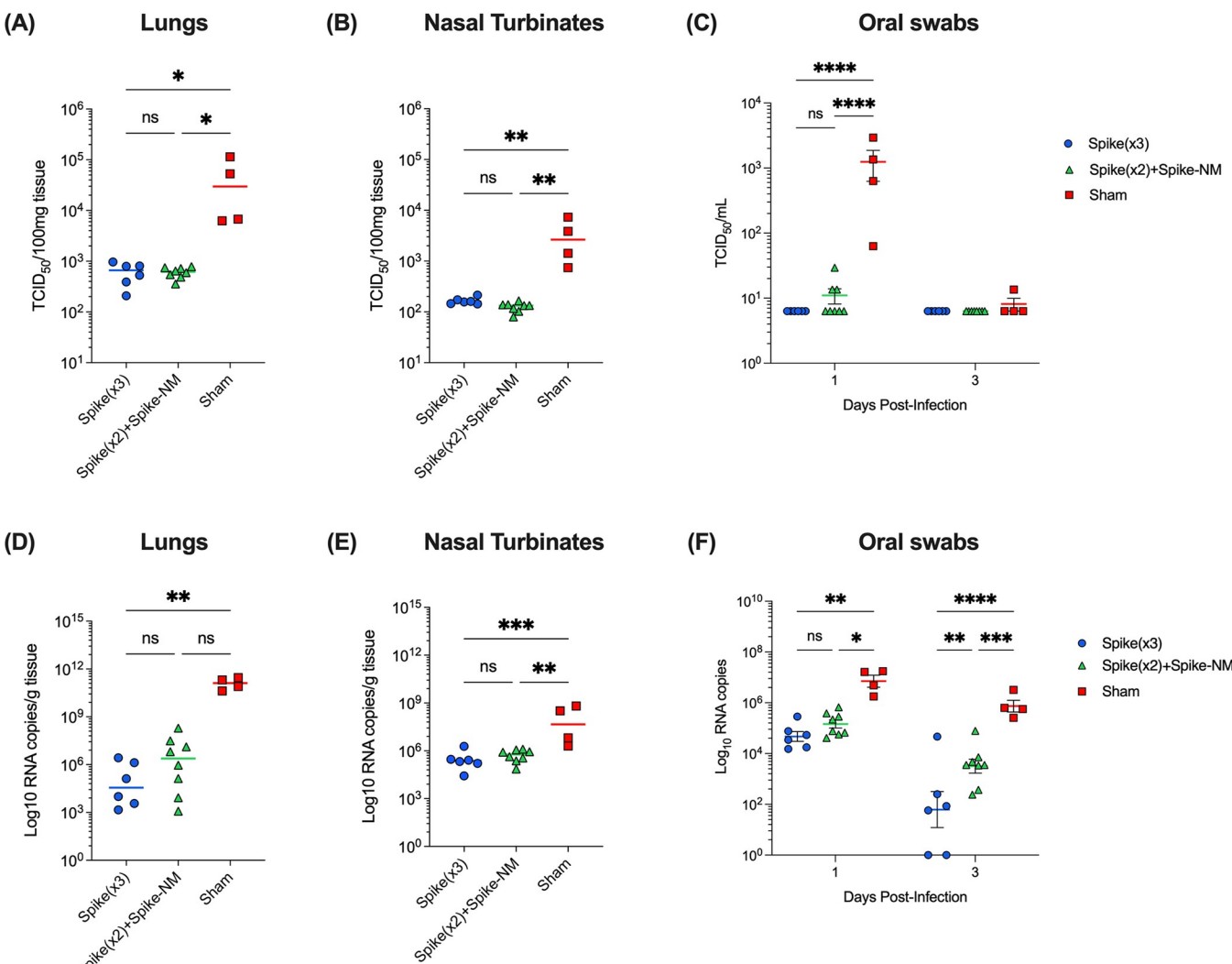

**Fig 5. Monovalent and multivalent booster vaccinations reduce viral shedding and viral RNA load in SARS-CoV-2 Omicron BA.5-challenged K18-ACE2 mice.** K18-hACE2 mice were intranasally challenged with $10^6$ TCID$_{50}$ SARS-CoV-2 Omicron BA.5 3 weeks following monovalent or multiantigenic boosters. From oral swabs, **(A)** amount of infectious virus and **(B)** SARS-CoV-2 E gene RNA copies were monitored at 1- and 3-days post-infection (dpi). At 5 dpi, mice were humanely euthanized with lungs, and nasal turbinates collected to determine whether vaccination reduces viral shedding and viral RNA loads. Amount of infectious SARS-CoV-2 virus are shown as TCID$_{50}$ per 100 mg tissue homogenates for **(C)** lungs and **(E)** nasal turbinates. Amount of SARS-CoV-2 RNA are shown as Log$_{10}$ RNA copies per g tissue for **(D)** lungs and **(F)** nasal turbinates. Each symbol represents one individual mouse (n ≥ 4 per group). For lungs and nasal turbinates, horizontal lines represent the median and statistical comparisons were made using one-way ANOVA with Dunn's or Tukey's multiple comparisons (*$p \leq 0.05$; **$p \leq 0.01$; ***$p \leq 0.001$). For oral swabs, horizontal lines represent the median ± IQR and statistical comparisons were made using two-way ANOVA with Tukey's multiple comparisons (*$p \leq 0.05$; **$p \leq 0.01$; ***$p \leq 0.001$; ****$p \leq 0.0001$).

## Cross-neutralization of SARS-CoV-2 VOCs

Sera was collected to assess humoral immunity at endpoint and to determine the effect on spike-specific neutralization when reducing the amount of spike mRNA from 10 µg to 5 µg and adding N and M to the booster dose. Using the pseudovirus assay, neutralization of SARS-CoV-2 Wuhan-Hu-1 (D614G), Beta (B.1.351), and Omicron (BA.1, BA.2, BA.2.12.1, BA.5, and XBB1.5) strains were measured from the sera of mice at 5 days following SARS-CoV-2 BA.5 challenge. As expected, both vaccinated groups elicited potent nAb levels against the matched pseudovirus harbouring the Beta B.1.351 spike while control mice receiving the formulation buffer only did not contain any neutralizing antibodies (Fig 7). Cross-

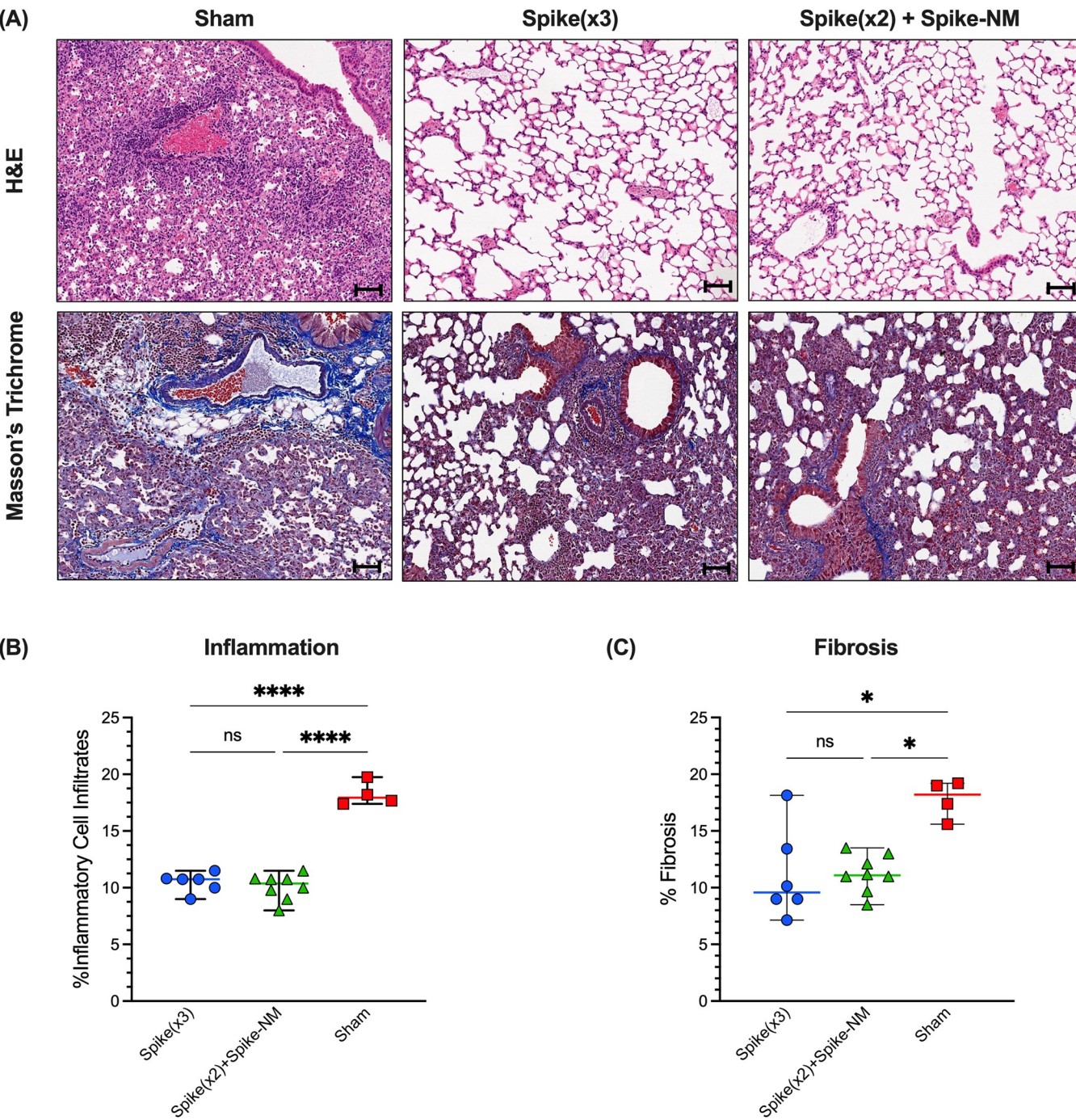

**Fig 6. Monovalent and multivalent booster vaccinations reduce lung pathology in SARS-CoV-2 Omicron BA.5-challenged K18-ACE2 mice.** K18-hACE2 mice were intranasally challenged with $10^6$ $TCID_{50}$ SARS-CoV-2 Omicron BA.5 at 3 weeks following monovalent or multivalent boosters. **(A)** Representative H&E staining and Masson's trichrome staining for lung pathology at 5 days post-infection (n ≥ 4 per group). Magnification is 20X with the scale bar representing 100μm. **(B)** Quantification of inflammation in lung tissues. Horizontal lines represent the median percentage of cellular infiltrates and error bars represent the range (n ≥ 4 per group). Statistical comparisons were made using one-way ANOVA with Tukey's multiple comparisons (*$p ≤ 0.05$; **$p ≤ 0.01$; ***$p ≤ 0.001$; ****$p ≤ 0.0001$). **(C)** Quantification of fibrosis (collagen staining) in lung tissues. Collagen is indicated by areas stained in blue. Horizontal lines represent the median percentage of fibrosis and error bars represent the range (n ≥ 4 per group). Statistical comparisons were made using Kruskal-Wallis test with Dunn's multiple comparisons (*$p ≤ 0.05$).

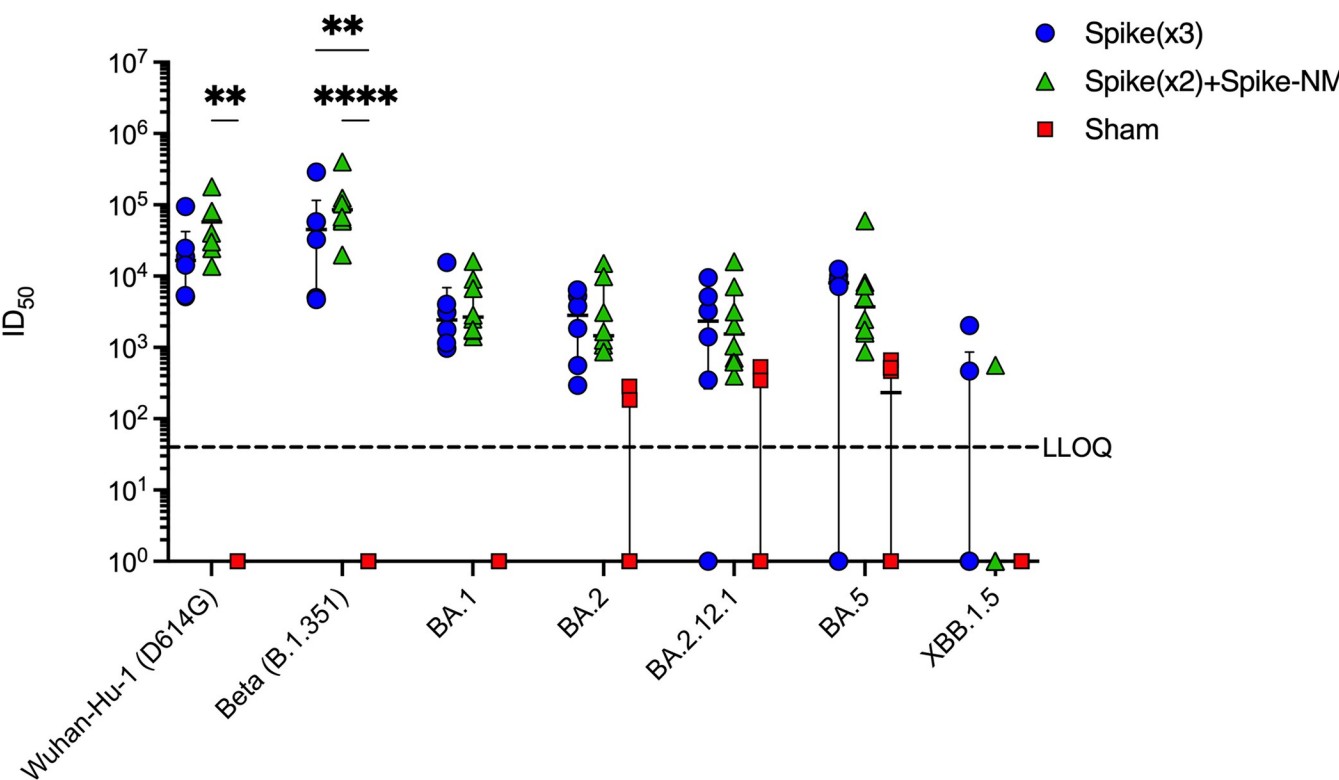

**Fig 7. Endpoint neutralizing antibody titers against SARS-CoV-2 VOCs from K18-hACE-2 mice following vaccination and SARS-CoV-2 Omicron BA.5 challenge.** Endpoint serum neutralizing antibody titers from mouse sera collected 5 days post-challenge with SARS-CoV-2 Omicron BA.5. Shown are neutralization of pseudoviruses bearing spike protein from SARS-CoV-2 Wuhan-Hu-1 D614G and VOCs, including Beta (B.1.351), and Omicron (BA.1, BA.2, BA.2.12.1, BA.5, and XBB1.5) strains. Each symbol represents $ID_{50}$ titers from one mouse ($n > 6$ per group). Samples that did not neutralize viruses at the lowest serum dilution (1:80 for XBB1.5, 1:40 for all other pseudoviruses) are designated an $ID_{50}$ titer of 1. The horizontal line indicates the median $ID_{50}$ titers for each group, and the short lines below and above the median indicate the 25th and 75th percentiles. Dotted line represents the lower limit of quantitation (LLOQ). Statical comparisons were made by two-way ANOVA with Dunnett's multiple comparisons test (*$p < 0.05$, **$p < 0.01$, ***$p < 0.001$, **** $p < 0.0001$).

neutralization against Wuhan-Hu-1 (D614G) spike was highest for the multiantigenic-boosted group compared to monovalent boosted sera, however, was not significantly higher than control sera (Fig 7). Omicron pseudoviruses bearing BA.1, BA.2, BA.2.12.1, and BA.5 spikes showed reduced neutralization potency from both vaccinated groups, and Omicron XBB1.5 pseudovirus showed the highest reduction to nAb activity, with median $ID_{50}$ titers at the lower limit of detection in the pseudovirus assay. In summary, our assessment of humoral immunity through the neutralization of pseudoviruses harbouring spike proteins of SARS-CoV-2 VOC revealed that the spike+NM multivalent booster produced higher neutralizing antibody levels against Beta (B.1.351) spike and cross-neutralization against Wuhan-Hu-1 (D614G) spike compared to the spike only booster. Serum reactivity against Omicron variants had reduced potency, with XBB1.5 showing the lowest reactivity to nAbs. These findings suggest that the inclusion of additional viral antigens in a multivalent vaccine supports similar neutralization of matched spike and cross-neutralization of spikes from VOCs as a monovalent vaccine containing spike only.

## Discussion

Nearly all currently approved COVID-19 vaccines utilize spike as the sole antigen and have been largely successful at preventing severe disease and hospitalizations, however, the rapid

emergence of SARS-CoV-2 variants that evade vaccine-induce immune responses and exhibit waning vaccine efficacy have underscored the need for second-generation vaccines that include more conserved antigens and induce more broadly protective responses. The multiantigenic booster approach provides an advantage over currently available vaccines by incorporating additional antigens relative to spike-only vaccines. Here, we investigated a booster immunization that incorporates the spike protein as well as a chimera containing nucleoprotein and membrane protein. We present preclinical findings that demonstrate the Beta spike-NM booster is as effective at preventing SARS-CoV-2 Omicron BA.5 virus shedding in K18-hACE2 mice as a monovalent booster containing Beta spike only. Our results showed that after BA.5 intranasal inoculation, high levels of viral titers were detected in oral swabs, lungs, and nasal tissues of unvaccinated mice, whereas mice receiving either homologous spike only or heterologous spike-NM booster had nearly complete inhibition of infectious virus shedding in oral swabs and reduced viral burdens in both lung and nasal tissues following BA.5 challenge. Examination of lung pathology further revealed that the spike-NM booster and the spike only booster provided comparable protection against lung pathology, including inflammatory infiltrates and fibrosis. Lastly, we also showed that that the breadth of neutralizing responses against the spike proteins of VOCs is similar between the spike-only monovalent and multiantigenic spike-NM boosters. Together, these findings demonstrate that the inclusion of additional SARS-CoV-2 antigens with spike in a booster immunization provides equivalent immunity and protection against SARS-CoV-2 Omicron BA.5 challenge to that of a booster vaccine that contains only spike.

The mRNA vaccine doses used in this study (5–10 μg) may be considered high relative to other studies that use 1–2 μg per dose [46, 47], however, vaccines containing 30 μg of mRNA have proven effective in mice without evidence of antibody-dependent enhancement of infection by virus-specific antibodies [48]. Furthermore, incremental benefits to immune responses can still be observed even at doses that are considered high [49–51], providing a more nuanced understanding of the mRNA dose-response relationship. The amount of spike in the heterologous spike-NM booster was decreased to 5 μg to maintain the overall dose of the spike-NM mRNA vaccine at 10 μg. Here, we demonstrate that this variation in the amount of spike in the third dose had no biological effect on protection against the BA.5 challenge or neutralizing responses, likely because all vaccinated mice received the same 10 μg dose of spike for their first two immunizations. Despite containing half the amount of spike mRNA as the monovalent spike-only, the spike-NM booster used in this study induced potent nAb titers that were comparable or non-inferior to the spike booster alone, including nAbs to a broad range of Omicron spike antigenic variants.

Although both boosters induced neutralizing antibodies against BA.1, BA.2, BA.2.12.1, and BA.5 in nearly all immunized mice, they were not significantly greater than control sera and antibody titers against XBB.1.5 were markedly lower with median values below the lower limit of quantitation. Previous studies have shown that mutations introduced in XBB.1.5 contribute to escape from neutralizing antibody responses in humans, and thus, the observed reduction to XBB.1.5 could be attributed to the additional substitutions found in the XBB.1.5 RBD (G339H, R346T, L368I, V445P, G446S, N460K, F486P, and F490S) compared with the BA.4/5 RBD [21]. For the current study, not all Omicron subvariant strains were available for use in a live virus neutralization assay at the time of testing, and thus, the pseudovirus assay was chosen to demonstrate that spike-based neutralization does not diminish when reducing the amount of spike mRNA from 10 μg to 5 μg and adding N and M to the booster dose. Pseudovirus neutralization assays have been utilized to demonstrate immunogenicity in the Pfizer-BioNTech and Moderna COVID-19 vaccine clinical trials [52–54], but no standardization procedures have been established as of yet. Furthermore, participants receiving a multi-epitope vaccine

containing peptides from S, N, and M proteins demonstrated strong correlations between pseudo- and live-virus neutralization experiments, according to a recent study by Wang *et al.* [25].

While most vaccines are based on sequences in the SARS-CoV-2 spike proteins, multiantigenic vaccines are a developing strategy for eliciting broader protection and the subject of recent studies. A phase 2 clinical trial testing a three-dose multi-epitope vaccine containing S1-RBD-sFc fusion protein enriched with peptides from N, M, and S2 proteins showed cross-neutralizing antibodies against WT (D614G), Delta, and Omicron strains, and T cell immunity that recognizes spike (S1-RBD and S2) and non-spike N and M proteins [25] but no vaccine efficacy data is available yet. Resch, *et al.* generated virus-like particle (VLP) vaccines to express spike, N, M, and E proteins and found high neutralization of VOCs and reduced viral loads following Beta and Delta challenges in Golden Syrian hamsters, however, they did not compare their multivalent VLPs to a spike-only VLP to determine any synergistic effects [55]. Cacciottolo *et al.* produced a bivalent protein-based vaccine delivered by exosomes to assess the immunity of exosomes-expressing spike in combination with N, and while animals were not challenged with SARS-CoV-2 to determine protection against virus load, they demonstrated strong immunogenicity of the spike+N exosome product with significant increases in antibody loads post-vaccination in both mice and rabbits [27]. Similarly, other groups have reported multivalent viral vaccine approaches co-expressing S and N on vesicular stomatitis virus (VSV)-based [56] or adenovirus-based [57] and found that the combination of S and N antigens broadens vaccine protection to the brain and lungs compared to the S only vaccine that protects against the lungs alone. However, O'Donnell *et al.* found that their dual-antigen VSV-based vaccine was protective as a single dose only when administered intranasally 28 and 10 days prior to viral exposure [56], and Dangi *et al.* only illustrated the reduction of SARS-CoV-2 RNA and not infectious virus titers or any pathological findings [57]. Overall, these studies are consistent with our findings that the addition of other non-spike targets results in similar or superior immunity and protection against SARS-CoV-2 VOCs and underscores the importance of further research into multiantigenic vaccine approaches. To our knowledge, our work is the first to show the effect of a multiantigenic mRNA booster following a complete prime and boost spike-only series that reflects the vaccination status of the majority of the global population.

Although immunogenic and cross-reactive epitopes identified in N and M proteins may be important for broad humoral and cellular immune responses critical for a broadly protective coronavirus vaccine [58–63], anti-N and anti-M antibodies were not specifically measured in this study. Comprehensive antibody analyses of COVID-19 patients demonstrate that N and M are immunogenic [64–68] and may have important roles in disease modulation outside of neutralizing antibodies. Using an ex vivo assay, a convalescent patient-derived monoclonal antibody targeting the N protein caused a conformational change when it bound to N, inhibiting MASP-2 hyperactivation and binding to the transcriptional regulatory sequence, ultimately preventing N protein-induced complement hyperactivation [69]. Since complement hyperactivation by SARS-CoV-2 can drive systemic inflammatory responses and lung injury in COVID-19 patients [70, 71], anti-N-protein antibodies may be important determinants for preventing aberrant complement activation during severe disease. This N-specific mAb also cross-reacted with the SARS-CoV N protein and the MERS-CoV N protein, highlighting the importance of generating N protein antibodies for a pan-coronavirus response. Furthermore, in studies of antigen-dependent Fc functions with SARS-CoV-2 antigens, Díez *et al.* show that antibodies in hyperimmune human convalescent plasma induced powerful antibody-dependent cellular phagocytosis (ADCP) and antibody-dependent cellular cytotoxicity (ADCC) activity for the N protein [72], suggesting that anti-N-protein antibodies may supplement neutralizing activity by enabling non-neutralizing antibodies or antibodies with poor-neutralizing

capacity to block or clear infection. Work by López-Muñoz *et al* supports this, showing that N protein is abundantly present on the surface of infected cells and can be targeted by Fc-expressing innate immune cells [73]. In contrast, the effectiveness of anti-M antibodies is less defined and may be influenced by sampling time. Some studies only occasionally detect anti-M antibodies in COVID-19 patients [67, 68], while others have detected IgM- and IgG-specific antibodies to the N- and C-terminal ends of the M protein in a significant portion of COVID-19 patients during the first weeks of COVID-19, with almost identical levels as observed for epitopes located in the spike and N protein [64]. To test the role of N or M-specific antibodies from vaccinated mice in this study, mouse sera could be depleted of antibodies against spike and the RBD before testing the neutralization capacity for the remaining antibodies and the effect on complement deposition. Future work should also test the cross-reactivity of SARS-CoV-2 N and M protein-specific antibodies among VOCs, as N-targeted antibodies in COVID-19 recovered patients are highly cross-reactive to Beta (B.1.351), Gamma (P.1), Delta (B.1.617.2), and Omicron (B.1.1.529) VOCs [74], suggesting that immunogenic epitopes within the N protein are not under selective pressure. Cross-reactivity against HCoVs should also be examined, as high levels of antibodies against the nucleocapsid of HCoV-OC43 have been associated with reduced probability of SARS-CoV-2 infection in healthcare workers in the Netherlands [75].

Prior studies that examined nucleocapsid-specific T cell epitopes in donors during active COVID19 and after recovery identified the HLA-B*07:02 restricted epitope, SPRWYFYYL, encoded by the N105-N113 peptide as the most immunodominant SARS-CoV-2 T cell epitope with no mutations observed in VOCs to date. This N105+ epitope leads to robust CD8+ T cell responses that can persist for up to 6 months after infection and demonstrates highly functional avidity and antiviral efficacy [36, 38, 76]. Furthermore, Tarke *et al*. and Heide *et al*. showed that N contains CD4+ epitopes N261-275, N306-320, and N336-350 [58, 77]. Therefore, confirming epitope-specific immune responses from the Beta spike-NM booster should be a future goal to validate cross-variant immunogenicity. However, any immunodominant N and M epitopes present could be HLA-restricted and unable to stimulate mouse T cells to clonally expand and function to promote antiviral efficacy in the K18-hACE2 mouse challenge model presented here. Thus, in circumstances where neutralizing antibodies are lacking or have been evaded by the VOC, vaccine efficacy should be confirmed with the VOCs as the challenging strain or by using a transgenic HLA-expressing mouse model.

To conclude, the present study demonstrates that incorporating N and M antigens with spike in a booster immunization provided equal efficacy as a spike-only booster at preventing severe disease by SARS-CoV-2 Omicron BA.5 challenge in K18-hACE2 mice, illustrated by comparable protection against virus load and shedding, prevention against severe lung pathology, and similar neutralization efficacies. In a reinvigorated global attempt to prevent SARS-CoV-2 spread, a vaccine that promotes immune responses to more conserved coronavirus proteins may be particularly relevant for variants capable of escaping anti-S neutralization and may prove to be a promising vaccination strategy for both naturally infected individuals and those who have already received vaccinations.

## Supporting information

**S1 File. This file includes S1-S16 Tables.**
(DOCX)

## Acknowledgments

We thank and acknowledge the full team at Providence Therapeutics for work on developing this vaccine and for the formulation of the vaccines used in the animal studies.

We thank Reuben Samson and Dr. Queenie Hu for developing the pseudovirus assay and generating some viral stocks, Dr. Jesse Bloom for sharing constructs, and Dr. W. Rod Hardy for cloning the spike VOCs into the lentivirus system. The generation of the spike VOC assays was supported by CoVaRR-Net, the Canadian Institutes of Health Coronavirus Variants Rapid Response Network. Anne-Claude Gingras is the Canada Research Chair in Functional Proteomics and a Pillar lead for CoVaRR-Net.

## Author Contributions

**Conceptualization:** Jacklyn R. Hurst, Robert A. Kozak.

**Data curation:** Jacklyn R. Hurst, Maedeh Naghibosadat, Patrick Budowski, Jun Liu, Anne-Claude Gingras, Natalia Martin Orozco, Robert A. Kozak.

**Formal analysis:** Jacklyn R. Hurst, Maedeh Naghibosadat, Patrick Budowski, Jun Liu, Philip Samaan, Frans Budiman, Alexandra Kurtesi, Fredo Qi, Haritha Menon, Anne-Claude Gingras, Mario Ostrowski, Natalia Martin Orozco, Robert A. Kozak.

**Funding acquisition:** Robert A. Kozak.

**Investigation:** Jacklyn R. Hurst, Maedeh Naghibosadat, Patrick Budowski, Jun Liu, Philip Samaan, Frans Budiman, Alexandra Kurtesi, Fredo Qi, Haritha Menon, Rajesh Krishnan, Anne-Claude Gingras, Mario Ostrowski, Natalia Martin Orozco, Robert A. Kozak.

**Methodology:** Maedeh Naghibosadat, Patrick Budowski, Jun Liu, Philip Samaan, Frans Budiman, Alexandra Kurtesi, Haritha Menon, Rajesh Krishnan, Jumai Abioye, Anne-Claude Gingras, Mario Ostrowski, Natalia Martin Orozco, Robert A. Kozak.

**Supervision:** Robert A. Kozak.

**Validation:** Anne-Claude Gingras.

**Writing – original draft:** Jacklyn R. Hurst.

**Writing – review & editing:** Jacklyn R. Hurst, Maedeh Naghibosadat, Patrick Budowski, Jun Liu, Philip Samaan, Frans Budiman, Alexandra Kurtesi, Fredo Qi, Haritha Menon, Rajesh Krishnan, Jumai Abioye, Anne-Claude Gingras, Mario Ostrowski, Natalia Martin Orozco, Robert A. Kozak.

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
