## [Decision Letter · Decision Letter 0]

5 Jun 2024

PONE-D-24-17590Comparison of a SARS-CoV-2 mRNA booster immunization containing additional antigens to a spike-based mRNA vaccine against Omicron BA.5 infection in hACE2 micePLOS ONE

Dear Dr. Kozak,

Thank you for submitting your manuscript to PLOS ONE. After careful consideration, we feel that it has merit but does not fully meet PLOS ONE’s publication criteria as it currently stands. Therefore, we invite you to submit a revised version of the manuscript that addresses the points raised during the review process.

We look forward to receiving your revised manuscript.

Kind regards,

Nagarajan Raju

Academic Editor

PLOS ONE

Journal Requirements:

   "This work was supported by a grant from the Canadian Institutes of Health Research"

   "J.L. and N.M.O are employees of Providence Therapeutics. This company was not involved in the funding of this research. J.L. and N.M.O were involved in some of the data collection and analysis as well as writing of the manuscript.

The other authors have declared that no competing interests exist"

We note that one or more of the authors are employed by a commercial company: Providence Therapeutics

5. In this instance it seems there may be acceptable restrictions in place that prevent the public sharing of your minimal data. However, in line with our goal of ensuring long-term data availability to all interested researchers, PLOS’ Data Policy states that authors cannot be the sole named individuals responsible for ensuring data access (http://journals.plos.org/plosone/s/data-availability#loc-acceptable-data-sharing-methods).

Additional Editor Comments:

I suggest authors to go through the comments from the reviewers and address them in the revised version of the manuscript

Reviewers' comments:

Reviewer's Responses to Questions

**Comments to the Author**

1. Is the manuscript technically sound, and do the data support the conclusions?

Reviewer #1: Yes

Reviewer #2: Yes

2. Has the statistical analysis been performed appropriately and rigorously? 

Reviewer #1: Yes

Reviewer #2: Yes

3. Have the authors made all data underlying the findings in their manuscript fully available?

Reviewer #1: Yes

Reviewer #2: No

4. Is the manuscript presented in an intelligible fashion and written in standard English?

Reviewer #1: Yes

Reviewer #2: Yes

5. Review Comments to the Author

Reviewer #1: The authors attempted to compare the immunogenicity and protective efficacy of a booster dose of mRNA encoding the S protein of the Beta variant alone or in combination with Wuhan M+N. Overall, the study design is sound and rational, and the strategy to answer the research question is scientifically valid. However, minor additional explanations are required:

1. The mRNA dose used in this study is relatively high (5-10 µg) in mice compared to other studies that used 1-2 µg per dose. Please discuss and compare the vaccine dose with those used in other studies. There is a concern that the high mRNA dose (5 or 10 µg) used in this study may obscure differences between treatment groups, as it could reach the plateau of antibody titers.

2. In writing, the authors should emphasize that the amount of S-mRNA in the spike-only group is double that in the multiantigen group. Thus, the efficacy contributed by anti-S antibodies may not be directly comparable.

3. Why is there no neutralization titer (Nt) data for the Beta variant in Figures 3A and 3B?

Reviewer #2: The manuscript of J. Hurst et al., ‘Comparison of a SARS-CoV-2 mRNA booster immunization containing additional antigens to a spike-based mRNA vaccine against Omicron BA.5 infection in hACE2 mice,’ is dedicated to assessing the potential value of SARS-CoV-2 nucleoprotein (N) and membrane (M) proteins in protection against new variants of SARS-CoV-2 infection.

The authors postulated that the vaccines expressing more conservative SARS-CoV-2 antigens, such as nucleocapsid and membrane proteins, might induce robust T cell immunity with a high frequency of cytokine-producing CD8+ T cells and be associated with a reduction of disease severity and/or a broader cross-neutralizing immune response.

The article is meticulously crafted, comprehensively describing the methods employed, key results, and a thorough review of the available literature.

Overall, the results are significant, indicating that the vaccine expressing anti-N and anti-M proteins is not associated with better spike-specific cell-mediated and humoral immunity (as expected) but also does not improve the prevention of heterologous SARS-CoV-2 infection in the transgenic mice challenge model. The N- and M-containing candidate has a lower impact on viral shedding than the monovalent anti-S comparator, induces comparable protection from severe lung pathology, and does not increase the breadth of neutralizing antibodies against heterologous SARS-CoV-2 strains.

The study results indicate that including more conservative epitopes in the vaccine composition does not translate into better protection or immunogenicity benefits in mice models.

In the discussion section, the authors try to justify the inclusion of N and M antigens, stating that they provide equal efficacy as a spike-only booster. However, the dose of mRNA used in the study (10 mcg) is relatively high for a booster, and 5 and 10 mcg can induce a similar immune response and be associated with a similar protection. As such, the hypothesis that the expression of anti-N and anti-M antibodies provides additional benefits is not supported by the data. Overall, the discussion overstated the value of the inclusion of N and M antigens, and the data do not support the conclusion.

Some important information is missing in the Methods section:

1. Immunogen design is not presented (stabilized vs. native spike?)

2. Is it modified or non-modified mRNA?

3. How was the expression of all target proteins controlled/verified? Was a potency assay used? Without measuring anti-N and anti-M antibodies, how did the authors confirm the sufficient level of expression of these antigens?

4. How did the authors select the dose level (10 mcg) for the mice experiments? If the intention is to demonstrate the value of N and M antigen inclusion, what was the rationale for reducing the S dose to 5 mcg?

5. Since a separate drug substance with N and M genes was produced, do the authors consider testing an anti-N/anti-M vaccine without spike protein to assess the stand-alone value of such vaccine composition?

The authors may refer to experience with inactivated COVID-19 vaccine, which broadens both non-spike-specific and spike-specific T-cell responses against SARS-CoV-2 (https://www.ncbi.nlm.nih.gov/pmc/articles/PMC10126277/) but was not associated with incremental vaccine efficacy (especially against new emergent variants; https://www.thelancet.com/journals/laninf/article/PIIS1473-3099(22)00732-0/fulltext)

6. PLOS authors have the option to publish the peer review history of their article (what does this mean?). If published, this will include your full peer review and any attached files.

Reviewer #1: **Yes: **Eakachai Prompetchara

Reviewer #2: No

---

## [Author Response · Author response to Decision Letter 0]

27 Aug 2024

Response to Editorial comments

We thank the editor and have ensured that the manuscript conforms to PLOS ONE’s style requirements

We have corrected this.

 "This work was supported by a grant from the Canadian Institutes of Health Research"

We have provided this statement in the cover letter.

 "J.L., H.M., R.K., J.Aand N.M.O are employees of Providence Therapeutics. This company was not involved in the funding of this research.H.M., R.K., J.A were involved in mRNA production, lipid nanoparticle formulation and quality tests of the mRNA vaccines. J.L. and N.M.O were involved in some of the data collection and analysis as well as writing of the manuscript. Commercial interests does not alter our adherence to PLOS ONE policies on sharing data and materials.

The other authors have declared that no competing interests exist"

We note that one or more of the authors are employed by a commercial company: Providence Therapeutics

We have amended this in the resubmission.

5. In this instance it seems there may be acceptable restrictions in place that prevent the public sharing of your minimal data. However, in line with our goal of ensuring long-term data availability to all interested researchers, PLOS’ Data Policy states that authors cannot be the sole named individuals responsible for ensuring data access (http://journals.plos.org/plosone/s/data-availability#loc-acceptable-data-sharing-methods).

We have amended this after discussion with all authors and Providence Therapeutics. 

All data available upon reasonable requests. 

Response to reviewers

Reviewer #1 Comments and Author Responses (written in bold). 

 The authors attempted to compare the immunogenicity and protective efficacy of a booster dose of mRNA encoding the S protein of the Beta variant alone or in combination with Wuhan M+N. Overall, the study design is sound and rational, and the strategy to answer the research question is scientifically valid. However, minor additional explanations are required:

We thank Reviewer #1 for their overall positive comments on the manuscript. 

1. The mRNA dose used in this study is relatively high (5-10 µg) in mice compared to other studies that used 1-2 µg per dose. Please discuss and compare the vaccine dose with those used in other studies. There is a concern that the high mRNA dose (5 or 10 µg) used in this study may obscure differences between treatment groups, as it could reach the plateau of antibody titers.

While the antibody plateau effect from high mRNA doses is a concern in some contexts, it is not universally observed across all studies and vaccine candidates. In the studies referenced below, the authors have shown that immune response to higher mRNA doses can vary, and incremental benefits can still be observed even at doses that are considered high. For example, in Corbett et al., nonhuman primates received 10 or 100 mcg of mRNA-1273. The higher doses of the mRNA-1273 vaccine did not show a simple plateau in antibody responses, and differences in antibody responses were still detected at the higher dose (Corbett et al., 2020, N Engl J Med, DOI: 10.1056/NEJMoa2024671). Additionally, in the phase 1 trial reporting of a dose-escalation of mRNA-1273, participants received two vaccine doses of either 25 mcg or 100 mcg of mRNA administered 28 days apart. Anderson et al showed that virus neutralization was dose-dependent, with responses to the 100-mcg dose higher than responses to the 25-mcg dose, and this neutralizing activity remained high through 4 weeks after the administration of the second dose in all the subgroups (Anderson et al., 2020, N Engl J Med, DOI: 10.1056/NEJMoa2028436). In Vogel et al., antibody responses characterized in BALB/c mice after a single intramuscular injection of 0.2-, 1- or 5-mcg show that 50% pseudovirus-neutralization titers were dose-dependent and were greater for the 5-mcg dose compared with the 1- or 0.2-mcg dose at 28 days post-vaccination. Furthermore, both binding and 50% pseudovirus-neutralization titers showed similar dose-dependent responses in rhesus macaques injected with 30 mcg or 100 mcg with geometric mean titers (GMTs) nearly double for the 100-mcg dose drawn 7 days after dose 2. By 21 days post-dose 2, titers for the 100-mcg dose remained at more than double the GMTs for the 30-mcg dose (Vogel et al., 2021, Nature, DOI: 10.1038/s41586-021-03275-y). Results from these studies demonstrate that higher doses continue to elicit robust antibody responses and that the immune system's response can still be dose-dependent even at higher mRNA doses without clear evidence of a plateau, providing a more nuanced understanding of the dose-response relationship. 

Nevertheless, when we reduced the amount of the spike in the heterologous spike-NM booster to 5 mcg to keep total dose of the spike-NM mRNA vaccine at 10 mcg, we did not observe any biological effect on the protection against the BA.5 challenge or neutralizing responses, likely because all vaccinated mice received the same 10 μg dose of spike for their first two immunizations. We have included the following statement on lines 539-551 to discuss how the high mRNA dose (5 or 10 µg) used in this study is unlikely obscuring differences between treatment groups, citing the studies above to describe that the more nuanced relationship with mRNA dose-responses: “The mRNA vaccine doses used in this study (5-10 µg) may be considered high relative to other studies that use 1-2 µg per dose (46,47), however, vaccines containing 30 µg of mRNA have proven effective in mice without evidence of antibody-dependent enhancement of infection by virus-specific antibodies (48). Furthermore, incremental benefits to immune responses can still be observed even at doses that are considered high (49-51), providing a more nuanced understanding of the mRNA dose-response relationship. The amount of spike in the heterologous spike-NM booster was decreased to 5 μg to maintain the overall dose of the spike-NM mRNA vaccine at 10 μg. Here, we demonstrate that this variation in the amount of spike in the third dose had no biological effect on protection against the BA.5 challenge or neutralizing responses, likely because all vaccinated mice received the same 10 μg dose of spike for their first two immunizations. Despite containing half the amount of spike mRNA as the monovalent spike-only, the spike-NM booster used in this study induced potent nAb titers that were comparable or non-inferior to the spike booster alone including nAbs to a broad range of Omicron spike antigenic variants.”

2. In writing, the authors should emphasize that the amount of S-mRNA in the spike-only group is double that in the multiantigen group. Thus, the efficacy contributed by anti-S antibodies may not be directly comparable.

We thank Reviewer #1 for their comment. The efficacy of the anti-S antibodies may not be directly comparable between the two vaccine groups as the group that received the heterologous spike-NM booster received half the amount of spike mRNA in their third dose (5 mcg) compared with the spike-only group that received 10 mcg of spike mRNA in their third dose. For the purpose of this study, we focused on making sure the heterologous booster was not inferior to the standard homologous spike-only booster and the results of the endpoint pseudovirus neutralizing titers confirmed this showing similar neutralization of the homologous Beta (B.1.351) strain (Figure 7). In lines 544-551 of the discussion section, we have included the text “The amount of spike in the heterologous spike-NM booster was decreased to 5 μg to maintain the overall dose of the spike-NM mRNA vaccine at 10 μg. Here, we demonstrate that this variation in the amount of spike in the third dose had no biological effect on protection against the BA.5 challenge or neutralizing responses, likely because all vaccinated mice received the same 10 μg dose of spike for their first two immunizations. Despite containing half the amount of spike mRNA as the monovalent spike-only, the spike-NM booster used in this study induced potent nAb titers that were comparable or non-inferior to the spike booster alone, including nAbs to a broad range of Omicron spike antigenic variants.”

3. Why is there no neutralization titer (Nt) data for the Beta variant in Figures 3A and 3B?

All data presenting immune responses to the Beta spike variant is shown in Figure 2, Immunogenicity of Spike B.1.351 in K18-hACE2 mice. In Figure 2C, serum neutralizing antibody titers to Beta spike are shown at 3 weeks and 8 weeks following the second vaccination. The text describing these results are under subheading “Spike B.1.351 vaccination induces strong spike-specific cell-mediated and humoral immunity in K18-hACE2 mice.”

In Figure 3, Pseudovirus neutralization of SARS-CoV-2 VOCs from vaccinated mice sera, we present the cross-neutralization against Wuhan-Hu-1 (D614G), and Omicron (BA.1, BA.2, BA.2.12.1, BA.5, and XBB1.5) spike variants that were not included in the vaccine formulation. The text describing these results are under subheading “Spike B.1.351 only vaccine does not induce broadly neutralizing antibody titers in K18-hACE2 mice.”

---

## [Decision Letter · Decision Letter 1]

5 Nov 2024

Comparison of a SARS-CoV-2 mRNA booster immunization containing additional antigens to a spike-based mRNA vaccine against Omicron BA.5 infection in hACE2 mice

PONE-D-24-17590R1

Dear Dr. Kozak,

We’re pleased to inform you that your manuscript has been judged scientifically suitable for publication and will be formally accepted for publication once it meets all outstanding technical requirements.

Kind regards,

Nagarajan Raju

Academic Editor

PLOS ONE

Additional Editor Comments (optional):

Reviewers' comments:

Reviewer's Responses to Questions

**Comments to the Author**

1. If the authors have adequately addressed your comments raised in a previous round of review and you feel that this manuscript is now acceptable for publication, you may indicate that here to bypass the “Comments to the Author” section, enter your conflict of interest statement in the “Confidential to Editor” section, and submit your "Accept" recommendation.

Reviewer #1: All comments have been addressed

Reviewer #2: (No Response)

2. Is the manuscript technically sound, and do the data support the conclusions?

Reviewer #1: Yes

Reviewer #2: Yes

3. Has the statistical analysis been performed appropriately and rigorously? 

Reviewer #1: Yes

Reviewer #2: Yes

4. Have the authors made all data underlying the findings in their manuscript fully available?

Reviewer #1: Yes

Reviewer #2: Yes

5. Is the manuscript presented in an intelligible fashion and written in standard English?

Reviewer #1: Yes

Reviewer #2: Yes

6. Review Comments to the Author

Reviewer #1: All comments have been addressed, and the concerns have been incorporated into the discussion. The manuscript is now recommended for acceptance for publication.

Reviewer #2: (No Response)

7. PLOS authors have the option to publish the peer review history of their article (what does this mean?). If published, this will include your full peer review and any attached files.

Reviewer #1: No

Reviewer #2: No

---

## [Editor Report · Acceptance letter]

21 Nov 2024

PONE-D-24-17590R1 

PLOS ONE

Dear Dr. Kozak, 

I'm pleased to inform you that your manuscript has been deemed suitable for publication in PLOS ONE. Congratulations! Your manuscript is now being handed over to our production team.

Kind regards, 

on behalf of

Dr. Nagarajan Raju 

Academic Editor

PLOS ONE